# Interaction Preference Differences between Elderly and Younger Exergame Users

**DOI:** 10.3390/ijerph182312583

**Published:** 2021-11-29

**Authors:** Ying Wang, Yuanyuan Huang, Junjie Xu, Defu Bao

**Affiliations:** Universal Design Institute, Zhejiang Sci-Tech University, Hangzhou 310018, China; 18800579952@163.com (Y.H.); junjie.xu@joynext.com (J.X.); defubao@zstu.edu.cn (D.B.)

**Keywords:** whole-body interaction, elderly users, exergame, preference difference

## Abstract

Existing motion capture technology can efficiently track whole-body motion and be applied to many areas of the body. This whole-body interaction design has gained the attention of many researchers. However, few scholars have studied its suitability for elderly users. We were interested in exercise-based whole-body interactive games, which can provide mental and physical exercise for elderly users. We used heuristic evaluation to measure participants’ actions during exergame tasks and analyzed preference differences between elderly and younger users through the distribution of actions in four dimensions. We found that age affected the actions performed by users in exergame tasks. We discuss the mental model of elderly users during the process of performing these tasks and put forward some suggestions for interactive actions. This model and these suggestions theoretically have guiding significance for the research and application of exergame design for elderly users and may help designers develop more effective exergames or other whole-body interaction interfaces suitable for elderly users.

## 1. Introduction

Following rapid development in the fields of motion capture technology and natural human–computer interaction technology, interaction methods of game playing based on whole-body movements have attracted widespread attention. Whole-body interaction allows users to interact with devices in a more intuitive and simple manner [1], making them feel more natural and comfortable than with traditional interaction methods [2,3]. Whole-body interaction technology has also been increasingly applied in the field of games, which brings immersive experience to the players and increases the fun of the game. Most of the research was conducted using Nintendo Wii. However, Nintendo Wii needs a handheld device to play, which may limit the player’s physical performance. Microsoft Xbox Kinect does not need any handheld devices to connect with video games. Players can play games through their own body movements, such as jumping, squatting, running in place and stepping to one side. This paper is based on Kinect to study the whole-body interaction. Bianchi-Berthouze [4] found that physical exercise enhances the user’s emotional and social experience of the game. Other authors have pointed out that exergames stimulate physical activity through upper-limb or whole-body exercise, which may help to improve health hazards caused by sedentary behavior [5]. It can also encourage and help elderly people to take physical and mental exercise, delay disease occurrence, maintain health and thus improve quality of life [6,7]. Therefore, among numerous whole-body interactive games, the exergame function that can help the elderly recover, maintain their health and improve their quality of life deserves further study.

Our cognitive and physical abilities decline with age. For example, perception acuity, memory and attention all decline [8]. As less muscle mass leads to a decrease in physical strength [9], exercise control ability declines [10], and physical damage may be caused by various cardiovascular diseases [11]. Additionally, it is difficult for elderly people to obtain information that enables them to engage with modern trends, meaning their execution actions during some games may differ from those of young people. The current game mechanism targeting young people is challenging and emphasizes players’ abilities. Thus, it is not suitable for elderly people. It is also inappropriate for this user group to follow the interactive gestures and design guidelines of the previous general environment [3,12].

Studies and extensive discussions have been carried out on these issues. Gerling et al. [13] proposed guidelines for the whole-body interaction of elderly people, including age-inclusive design, fatigue management, dynamic difficulty, simple gesture recall and continuous assistance. However, many interactions remain unnatural, and elderly users may have difficulty learning and remembering them [1,14]. This may be because these interactions are defined by designers.

We therefore conducted a comparative experimental study on exergames, based on whole-body interaction, which provides mental and physical exercises for the elderly. We used a set of classification methods suitable for interactive actions in exergames to classify participants’ task actions. Through further analysis, we explored preference differences between elderly and general users during whole-body interaction. On this basis, we propose a psychological model of elderly users when performing exergame tasks and provide suggestions for the design of whole-body interaction actions for elderly users.

The purpose of this study is to compare the differences of action preferences between young people and the elderly, further analyze the mental model of the elderly subjects and put forward some suggestions on action design of exercise games for the elderly.

The rest of this article is organized as follows: In Section 2, we review work related to game design and exergames for elderly people and action classification. In Section 3, we describe our controlled experimental process, data processing and analysis. In Section 4, we present the experimental results. In Section 5, we discuss the causes of the results, propose psychological models and put forward design suggestions. In Section 6, we summarize the contributions of this study and possible future research.

## 2. Related Work

The cognitive and ability of elderly exergame users have certain restrictions. First, we describe the current situation and significance of game gesture design, based on the needs of elderly people. Second, we explore the differences in the actual research of action classification and how to build on the existing work. Finally, we provide an overview of exergames and look at their application, as well as reviewing related research on the motives and preferences of elderly people when playing games.

### 2.1. Research on the Design of Game for Elderly People

Game design for elderly people can be traced back to the 1980s. Weisman [15] explored the usability of Apple II games for elderly people in institutions, highlighting the importance of adaptive games. Many scholars have since carried out relevant research to solve the problem of adaptability. For example, Ijsselsteijn et al. [16] proposed a set of designs with individual meaning and visually adjustable multi-modal feedback by analyzing age-related changes. This meaningful game design was intended to emphasize the importance of benefits for elderly people. Flores et al. [17] researched games created for stroke rehabilitation and suggested visual adaptability, including appropriate therapeutic exercise and cognitive challenges. Aarhus et al. [18] highlighted the importance of allowing elderly users to adjust game speed and difficulty to adapt to individual differences. Gerling et al. [13] pointed out that age-related injuries affect elderly people playing games in many ways and proposed suggestions for older people in institutions to solve the usability problem of balance exergames. Most authors have focused on analyzing the needs of elderly people and designing game mechanisms based on them. They have given high-level suggestions for game design for elderly people, but the design of interaction between elderly people and the system is also worth exploring deeply.

In 2009, authors pointed out that game design can operate or interact with devices with commands through the interface [19]. By 2013, gesture field had become one of the most important research topics in human–computer interaction (HCI) [20]. Some scholars have pointed out that gestures are intuitive, easy-to-use, do not need prior training and interact with technology quickly, meaning they can encourage elderly people to use them [21]. Gesture-based interaction has a history in the field of 3D user interface design [22,23]. However, elderly people have difficulty executing the interactive gestures, and the original whole-body game gestures were not always suitable for this group [24].

### 2.2. Exergame

The positive significance of exergames for elderly people deserves an in-depth study. As a way to improve users’ health through exercise, the term ‘exergame’ was included in the Collins English dictionary in 2007. Many authors have analyzed the mechanism and effects of exergame. For example, Van et al. [25] reviewed 13 research papers on the effect of exergames on balance training, summarized the effect of exercise training on the posture control of elderly people and proposed a low-cost sensing system to quantify posture control ability as it changes during the game, continuously monitoring balance ability without the user’s awareness. Based on this, Van et al. generated exergame training plans and feedback for individual users.

Additionally, many researchers have researched the motives and preferences of elderly people when playing games. Cota et al. [26] established a game manual to determine elderly people’s game preferences, developed a game prototype and combined post-event interviews to explore elderly game players’ motivations. Frey and Osterloh [27] found that the biggest motivational factor for elderly people to play and keep playing games was understanding the benefits games can provide. This offers a basis for the existence of whole-body interactive exergames that can benefit the cognition and motor abilities of elderly people. That is, games can motivate elderly people to exercise [28]. Therefore, we took exergames as an example for deeply exploring the preference differences of interactions between elderly and general users.

### 2.3. Classification of Actions

Efron [29] conducted one of the first studies of discursive human gesture resulting in five categories on which later taxonomies were built. The categories were physiographics, kinetographics, ideographics, deictics and batons. The first two are lumped together as iconics in McNeill’s classification [30]. McNeill also identifies metaphorics, deictics and beats. Because Efron’s and McNeill’s studies were based on human discourse, their categories have only limited applicability to interactive surface gestures. Poggi [31] further offers a typology of four dimensions along which gestures can differ: relationship to speech, spontaneity, mapping to meaning and semantic content.

Working on a pen gesture design tool, Long et al. [32] showed that users are some-times poor at picking easily differentiable gestures. To address this, Wobbrock et al.’s [33] guessability methodology resolves conflicts among similar gestures by using implicit agreement among users. Each gesture is manually classified according to four dimensions: form, nature, binding and flow. The scope of the form dimension is within one hand. It is applied to each hand with two-handed gestures. The interactive actions studied in this paper are not limited to hands but also include trunk and leg movements. Therefore, based on the above research, this paper creates a clear multi-dimensional classification method for whole-body interactive actions.

## 3. Materials and Methods

### 3.1. Participants

A total of 25 participants were recruited from a university and its surrounding communities, including 12 people aged 20–30 (M = 24.5, SD = 0.76) and 13 people aged over 60 (M = 65.2, SD = 4.25). The ratio of male to female in both groups was approaching 1:1. All participants’ eyesight or corrected eyesight was normal; they had no physical disorders, were all right-handed and could stand normally for at least 30 min to complete the experiment (Figure 1). Table 1 shows the characteristic information of the participants. To prevent the game task from producing cognitive impairment or mental stimulation which influenced the experiment, all participants had to pass the mini mental state examination (MMSE) [34], meaning they were mentally healthy. Before the experiment, all participants signed informed consent forms.

### 3.2. Game Tasks

After obtaining various types of exergames through market research, we evaluated and screened the collected games based on the relevant literature. In this way, we determined the types of exergames to include in this study. These exergames were a driving game for cognitive training and a sports game that exercised physical and balance ability. These games cover most of the existing forms and common operation tasks of exergames and can be used as a representative beginning to explore the action design of exergame. By extracting and defining the common operation commands of these games, we selected 20 basic tasks (Table 2): six from the driving game (drive, turn left and right, accelerate, shift, brake), nine sports game tasks (climb, jump, kick the ball, play table tennis, shoot, ski forward, ski left and right, ski slow down) and five system commands (call menu, zoom in, zoom out, select menu item, close menu).

### 3.3. Experimental Equipment

To prevent the influence of social effects, this experiment was conducted in a quiet indoor environment. The experimental equipment mainly includes: a notebook computer (MacbookPro2015, operating system is IOS10.13.6, screen size resolution: 15 inches, 2880 × 1800 resolution), which is used to play the prepared task material display; a projector (Sanyo) and projection screen, which enlarges the screen on the computer and projects it onto the projection screen; an SLR camera (Nikon D5000) was used to record the experimental video from the front side of the participant, which is convenient for subsequent identification and analysis; several scoring scales and pencils printed on A4 paper were used to record the basic information and subjective scores of participants.

### 3.4. Experimental Procedure

Before the experiment began, two experimenters arrived at the site. Experimenter A was responsible for preparatory work, including receiving participants, distributing the basic information questionnaire for participants to complete and initializing the equipment. Experimenter B explained the contents and precautions of the experiment to the participants, obtained informed consent and evaluated participants’ mental states.

After the experiment began, heuristic evaluation, as used by Wobbrock et al. [33], was adopted. First, instructions were presented to the participants on a large screen. The process and requirements of the experiment were then carefully explained to the participants, who were told to think aloud when performing gestures. The actions they performed were confirmed, and the corresponding basic principles described. We then reminded participants to assume that they were in a game environment and that they could use any body part to perform actions and could reuse actions. In this way, we could identify the preferred actions (Figure 2).

After the participants were ready, the formal experiment task interface was displayed on the large screen; that is, each experimental task appeared together with the task name and target scene. The target scene was two pictures or video clips of different target states before and after performing the task. Following this, the experiment host described the task target and scene to help participants understand the task. Participants were required to perform actions to cause changes in the target scene. They were about 1.5 m away from the monitor when performing the action, thus ensuring that there was enough space to act.

After all tasks were performed, a brief interview was conducted to inquire about and confirm the problems found in the experiment. The participants were asked to evaluate the actions they performed in the following three statements: “the actions I performed matched their purposes very well”; “The actions I perform are easy to perform”; and “I’m tired from the actions I perform”. All the evaluations were scored by a 7-point Likert scale, with 1 being “very different” and 7 being “very agree”. This was recorded via audio and video for analysis. The whole experiment lasted approximately 20 min (Figure 3).

### 3.5. Data Analysis and Processing

The amplitude and position of the actions varied from person to person, resulting in great differences in actions proposed by each person. Thus, it was difficult to analyze and compare actions directly. We therefore conducted action classification before further analysis. Actions were classified according to the audio recording and participants’ oral description, where actions with the same or similar motion or posture were regarded as the same. The action mainly includes the direction of joint movement (combined with the oral description of participants, there are mainly three movement directions: vertical, horizontal and oblique) and the synchronous movement of body parts. Posture refers to the ending position of all involved body parts moving. In some cases, participants essentially used the same action to execute specific commands but used different hands or feet, but we classified these actions as the same.

After classifying the actions of the subjects, we used a chi-square test to analyze the statistical differences of the actions of the two groups in the four classification dimensions due to the large number of classification dimensions and different groups (due to the presence of theoretical frequencies, less than 1 and 1 ≤ T< 5 of the number of grids exceeds 1/5 of the total number of grids, and part of the analysis used Fisher exact test). We need to pay attention to Cramer’s V coefficient, which ranges from 0 to 1. The larger the value, the stronger the correlation. In addition, we also paid attention to the adjusted standardized residuals: when the adjusted calibration residuals of the absolute value is greater than 2, we think that there is a significant relationship between the observed frequency and the expected frequency. In this experiment, due to the multiple comparisons involved, in order to estimate more conservatively, we took the absolute value equal to 3 as the boundary.

## 4. Results and Analysis

### 4.1. Classification Dimension of Actions

In total, 440 effective actions of 22 participants were obtained (three elderly participants did not complete the experiment). Wobbrock [33], Silpasuwanchai [24], and Chen and others [35] studied two-dimensional gestures, which usually involved direct contact with the interface, while the previous three-dimensional gesture research mainly focused on gestures or restricted body interaction research, while our body interaction research was unrestricted body movements, and the selected experimental tasks were game tasks, which led to higher movement changes. Based on the classifications of Wobbrock [33], Silpasuwanchai [24], and Chen and others [35], we proposed a classification method suitable for interactive actions in exergame, which has four dimensions: form, characteristic, position and body part. Each dimension has multiple categories (Table 3).

The form dimension represents the active form of an effective action. The static state involves simply moving a body part to a specific posture and holding it for a period before retracting. For example, the hands are claw-like to show an enlargement. Coherence refers to a set of coherent movements of one specific body part. For example, turning the arm to the left means turning left. Reciprocate refers to multiple reciprocating movements of specific body parts. For example, squatting and waving arms continuously means skiing forward.

In the characteristic dimension, reality mapping represents actions performed based on real-world objects, such as kicking the ball, using the steering wheel operation to represent driving, and clicking on a screen in the environment to represent the call menu. Metaphorical action means that the goal is to act as other similar objects. For example, the action of opening the scroll indicates opening, while clicking the palm indicates calling the menu. When there is no symbol, the reality mapping or metaphorical relationship between the action and the task goal, or if the action mapping of a subject is arbitrary, this is regarded as an abstract action. For example, raising your hands and flapping forward means acceleration.

The position dimension describes the relative position when performing an action. Object-centric means that an action acts on a specific object in the interface. For example, pressing the steering wheel in the interface with one hand means braking. Interface-dependent actions are related to interface definitions. For example, according to the interface environment, swinging the right hand back and forth at the side indicates gear shifting. Interface-independent ignore the characteristics of the interface and can be executed anywhere, for example, clicking a finger to indicate a call menu. Mixed-dependent actions mean that the action depends on the environment on one side and either depends on the object or is independent of environment on the other. For example, when one hand holds the steering wheel in the interface environment and the other presses the steering wheel, this indicates acceleration.

The body-part dimension counts the body parts that are most involved in the movement. These are divided into one hand, both hands, one leg, both legs and trunk. Combined actions involve at least two body parts, while other actions involve the whole body.

We further classified each dimension to observe differences in the distribution of actions performed by elderly and younger users in each dimension, thus exploring preference differences. Next, we describe this distribution in detail.

### 4.2. Preference Differences in Action

Figure 4 and Figure 5 show the percentage of all actions in our study, in each dimension. In the young group, coherence (75.83%), reality mapping (73.33%) and interface-dependent (71.25%) were much higher than other classifications in their respective dimensions, while both hands (30%) were the most popular in the body-part dimension. In the elderly group, the highest actions in each dimension classification were also coherence (62%), reality mapping (53%) and interface dependent (55%), but proportions were slightly decreased. In the body-part dimension, both hands (31%) were still the most popular, followed by one hand (29%), one leg (15%), whole body (10%), combination (8%), both legs (6%) and trunk (1%).

In the form dimension, the percentage of reciprocate actions in the elderly group was higher than in the young group (21.25%). In the characteristic dimension, the percentage of metaphorical (34.5%) and abstract (12.5%) actions in the elderly group was higher than in the young group (22.08% and 4.59%, respectively). In the position dimension, the elderly group’s interface-independent (17.5%) was higher than in the young group (4.58%).

To compare whether the two groups of participants had significant differences in various dimensions, we used a chi-square analysis and Fisher exact test (Table 4). After examination, in the form dimension, X^2^(2, N = 440) = 10.096, *p* < 0.05, indicating a significant preference difference in the actions of different groups in this dimension. There was a weak correlation between the classification of different groups and the form dimension (Cramer’s V = 0.151 < 0.3). Additionally, in combination with the adjusted normalized residual difference value, we found statistically significant differences between the young and elderly groups in the classifications of coherence and reciprocate. That is, although both the young and elderly groups preferred coherent action, the elderly group had a significantly lower preference for coherent action and a significantly higher preference for reciprocate actions than the young group.

In the characteristic dimension, X^2^(2, N = 440) = 21.460, *p* < 0.05, indicating a significant preference difference in the actions of different groups. There was also a weak correlation between the different groups and the classification of the characteristic dimension (Cramer’s V = 0.221 < 0.3). Additionally, in combination with the adjusted normalized residual value, we found that the difference between the two groups in reality mapping and abstract was statistically significant. While both groups preferred reality mapping actions, the elderly group had a significantly lower preference for reality mapping and a significantly higher preference for abstract.

In the position dimension, *p* = 0.000 < 0.05, indicating a significant preference difference in the actions of different groups for this dimension. There was a weak correlation between the classification of different groups and the position dimension (Cramer’s V = 0.226 < 0.3). Additionally, in combination with the adjusted normalized residual value, we found that the difference between the young and elderly groups was statistically significant for the classification of action between interface dependent and interface independent. Although both groups preferred actions that depended on the interface, the elderly group had a significantly lower preference for interface-dependent actions and a significantly higher preference for interface-independent actions.

In the body-part dimension, *p* = 0.483 > 0.05, indicating no significant preference difference in the actions of different groups in that dimension. Both groups preferred to use both hands to perform actions, and there was no statistically significant preference difference between the two groups on the various other body parts.

An analysis of the results and interviews was carried out. In the form dimension, it may be that the elderly group preferred reciprocating motion more than the young group due to the lack of feedback on certain tasks. Due to the decline of cognitive ability, elderly participants could not perceive the completion of action input and thus perform actions repeatedly, while the younger participants had clearer cognitions on the completion of an action. Another possible reason is that elderly people performed actions repeatedly to strengthen psychological confirmation due to uncertainty about whether those actions were correct.

In the characteristic dimension, elderly participants may have preferred metaphorical and abstract actions more than the younger participants as they chose relaxed actions for completing tasks, due to their reduced physical fitness. For some tasks, elderly people could not refer to realistic actions as they had no relevant experience. Therefore, they linked the task with other operations and performed metaphorical and abstract actions. This may also be why elderly participants preferred interface-independent actions in the position dimension compared to the younger participants.

Due to the characteristics of exergame, the body parts used to perform actions were related to the types of game tasks, and due to the small sample size, no significant difference was found between the two groups on this dimension. However, the elderly group seldom used whole-body movements, which may have been due to their physical limitations.

### 4.3. Difference of Action Distribution

We also found that for different tasks, the action distribution of the two groups in each dimension differed. Therefore, we further explored these differences in each game task in the four dimensions.

Figure 6 shows the action distribution of the two groups in the form dimension for each task. In the younger group, in addition to the two tasks of driving and climbing, which tended to reciprocate, the remaining 18 tasks showed the trend of using coherent actions. We also found that elderly participants tended to reciprocate in driving and climbing and showed a reciprocating trend in table tennis, skiing forward, zooming in and zooming out. This may be because the cognitive ability of the elderly people was declining; therefore, they could not accurately understand tasks and quickly identify feedback. It is worth noting that 50% of the right-skiing tasks were reciprocating, while the left-skiing tasks only accounted for 30% of the reciprocating motions. This was also reflected in the younger group, which may be because the participants in this experiment were all right-handed.

Figure 7 shows the action distribution of two groups’ participants in the characteristic dimension in each task. In the young group, the participants preferred to use metaphorical actions for shooting, zooming in, zooming out and closing menu, while for calling menu, the participants preferred to use abstract actions, and the other actions showed realistic mapping. In the elderly group, participants preferred metaphorical actions for shooting, skiing and calling menu. Additionally, for other tasks except braking, climbing, kicking a ball, playing table tennis, calling menu, selecting menu items and closing menu, the preference of abstract action in the elderly group was higher than in the young group. This may be because elderly participants preferred to use easier actions. Elderly participants had no relevant experience for specific tasks and so could not refer to realistic actions, meaning they preferred metaphorical and abstract actions. 

Figure 8 shows the action distribution of two groups’ participants in the position dimension in each task. Younger participants preferred to use object-centric actions for climbing, kicking, playing table tennis and selecting menu items. Notably, 75% of call menu tasks were interface-independent actions. In the elderly group, in addition to climbing, kicking, playing table tennis and selecting menu items, participants also preferred to use object-centric actions for zooming in and zooming out. In addition, in other tasks, the proportion of preference for interface-independent actions in the elderly group was higher than in the young group. This may be because elderly participants had no relevant experience in most tasks.

Figure 9 shows the action distribution of two groups’ participants in the body-part dimension in each task. For driving tasks, participants tended to use both hands, one hand and one leg. For skiing activities, participants tended to use combined actions and whole-body movements, while for menu operation tasks, participants tended to use one hand or both hands. Therefore, the body parts used by the participants are mainly related to the task objectives.

### 4.4. Summary

Compared with the young participants, the elderly participants preferred to use reciprocating, metaphorical, abstract and interface-independent actions, and fewer whole-body actions.

In the form dimension, elderly participants preferred a reciprocating motion more than the young participants. Perhaps the decline of their cognitive ability meant they could not perceive whether the task was completed and repeated, to strengthen the psychology of confirmation. In the characteristic dimension, elderly participants preferred metaphorical and abstract actions compared to the young participants, perhaps because physical actions and thinking constraints resulted in a preference for easier actions associated with life experience and real actions for performing tasks. If players have no relevant experience, they may link the task to other operations and perform metaphorical and abstract actions. This is also why the elderly group preferred interface-independent actions in the position dimension more than the young group. In the body-part dimension, there was no significant difference between the two groups. Less use of body parts in the elderly group may be due to the limitations of physical ability.

## 5. Discussion

### 5.1. Classification Method of Whole-Body Interactive Actions Suitable for Exergames

This paper puts forward a set of classification methods for the whole-body interactive actions of exercise games, in which the actions are classified into four dimensions: form, characteristics, position and body parts. On this basis, the differences of action preferences between young people and old people in various dimensions are analyzed. Advantages: The research of Wobbrock et al. [33] is a two-dimensional gesture, which usually involves direct contact with the interface, while the previous three-dimensional gesture research mainly focused on gestures or restricted whole-body interaction research. While our research on whole-body interaction is an unrestricted whole-body action, and the selected experimental task is a game task, which leads to a high movement change. Disadvantages: In this study, we only considered the standing interaction using only the whole body, while in the next research, we plan to introduce more body posture interaction, for example, studying the whole-body action design or the whole-body action with additional equipment when playing driving games in sitting position.

In the formal dimension, the preference of the elderly group for reciprocating motion is higher than that of the young group. On the characteristic dimension, the preference of the elderly group for metaphor and abstract action is higher than that of the young group. In the position dimension, the elderly have a higher preference for environment-independent actions than the young group. No significant difference was found in the dimension of body parts. In addition, in different tasks, the action distribution of the two age groups is also different. Compared with the young group, the old group shows a higher preference tendency of reciprocating, metaphorical abstraction and environment-independent action in more tasks.

Compared with other user-defined studies [33,36,37], the conclusions of our study in different dimensions are more diversified. This may be caused by the following reasons: (1) The research of Wobbrock et al. [33] is a two-dimensional gesture, which usually involves direct contact with the interface, while the previous three-dimensional gesture research mainly focused on gestures or restricted body interaction research. While our research on body interaction action is an unrestricted body action, the selected experimental task is a game task, which leads to a high movement change. (2) Previous studies mainly focused on the touchscreen interaction or targeted young people who have some knowledge of the whole-body interaction. For the whole-body interaction game, most of the subjects lack the corresponding experience, especially for the elderly, which leads to higher unknowns and uncertainties of the actions performed.

Before the start of the study, the author designed the corresponding execution actions for these actions in advance when choosing the test tasks, but the final result showed that the actions envisioned by the author only accounted for 11.11% of the actions performed by the subjects, and 20% of the actions designed by the author were never tried by the participants. This result is consistent with the previous research [33], which once again emphasizes the importance of bringing users into the design system.

### 5.2. Psychological Model

Through the analysis of the participants’ vocal thinking and experimental results, we have captured psychological models of elderly users when selecting and executing corresponding tasks and actions. These are reverse thinking, simplifying and connecting with reality, respectively.

#### 5.2.1. Reverse Thinking

Some examples of mirror tasks include turning left/right, skiing left/right, zooming in/out and so on. For these tasks, participants usually use reversible actions, that is, actions that produce the opposite effect in the opposite direction. This kind of user behavior affects the development and use of paired reversible actions for mirror task gestures in subsequent research. which, in turn, reduces the difficulty of system recognition.

Additionally, we found differences in participants’ actions to define mirror tasks. We preliminarily speculated that the reason for this was the left–right difference caused by the right-handed participants in this study. In the experiment, the sensational thinking of elderly participants who showed a difference between left and right was as follows: “It seems that turning right is more difficult than turning left”. This indicates that when users encounter a task with high complexity and so feel embarrassed (for example, the conceptual complexity of the left/right ski task is three, meaning this is a difficult task), the action performed is greatly influenced by the dominant hand.

#### 5.2.2. Simplifying

In the process of completing tasks, elderly participants often had simplified conjectures. For example, when a participant was asked to perform a zoom-in task, she said: “This is the same as zoom out, click”. She then performed the same action as zooming in the right index finger to click the imaginary button in the air. This led to similar actions in acceleration/brake, zoom-in/zoom-out, and select menu items/call menu/close menu, which allowed us to eliminate ambiguity by judging the position of the executed actions when designing action sets for elderly people.

#### 5.2.3. Connecting with Reality

The connection with reality exists in two situations. First, when participants perform familiar tasks, they usually perform actions according to reality. For example, the elderly participants in this experiment had experience of playing table tennis. Therefore, when performing table tennis tasks, most performed the right-hand swing from back to forth on the right side of the body, with a high consistency of actions. Additionally, when participants found it difficult to think of an executive action for a specific task, they tended to associate the task with a familiar real-life operation. For example, when performing a call menu task, most participants choose to click an imaginary button in the air or make an action of pressing the remote controller, in which their vocal thinking when executing the action included: “I turned on the TV to do this”, “I will click here” and so on. This indicates that participants associated the task of the call menu with the real-life operation of turning on a TV.

### 5.3. Suggestions on the Design of Exergames for Elderly Users

By summarizing the experimental results and analyzing the psychology of elderly people, we have put forward some suggestions on exergame action design for elderly users.

1. Inclusion of aging characteristics;

We must pay attention to the decline of cognitive and physiological abilities related to aging. The decline of cognitive abilities such as memory and attention affect the human–computer interactions of elderly people. These factors must be considered when designing interactive actions, evaluating the applicability of interactions and minimizing the risk of injury caused by inappropriate sports. For skiing forward, we suggest using the action of “leaning forward and row backward with both hands”; for skiing to the left/right, it is recommended to use the action of “leaning to the left/right front”; for skiing slow down, it is recommended to use the action of “stretching your body straight with your hands on both sides”.

2. Higher fault tolerance;

For elderly people, game motion recognition should have higher fault tolerance rather than a higher accuracy. Elderly people are affected by the reduction of exercise range, limiting their whole-body interactions. We found that for certain tasks, while all elderly participants like to perform with similar actions, based on their preferences and physical abilities, they differ in details such as displacement and starting position when performing actions. For example, to kick a ball, the participants performed “right foot kick forward” with different heights, angles and initial positions. Therefore, recognition systems should tolerate such individual differences. The customization of operating settings provided by the game system may help solve this problem. For shooting, it is difficult to aim. Therefore, we suggest to distinguish between the two scenes. In the entertainment scene where shooting accuracy is not high, we suggest using the action of “swing forward with the right hand as a dart projection”. If there are certain requirements for shooting accuracy, we suggest using the action of “shooting with the right hand as a pistol and lift the wrist lightly”.

3. Clear feedback;

Clear feedback should be provided after performing actions that affect the game scene. Elderly people’s attention was significantly decreased, meaning they often could not feel slight changes. For example, in this experiment, elderly participants showed a high tendency for reciprocating actions, which resulted in repeated actions as they could not feel whether the actions had been inputted or not. This shows the importance of providing clear feedback after elderly people perform actions. This is also helpful for providing a sense of immersion and improving the enthusiasm of elderly people for playing games.

4. Fatigue management;

Elderly people’s fatigue should be considered when designing exercises. Due to the decline of physical function, elderly people often have lower endurance levels and are more vulnerable to sports injuries and overwork. Games must manage players’ fatigue and guide elderly people to perform actions through the control of the game rhythm to achieve sustainable exercise and avoid excessive fatigue. For example, game phases alternating tension and relaxation allow players to exercise and rest. This means that action designs for elderly people should focus on the fatigue and physiological load caused by the action. If the game is played in a short time, actions that can produce a better exercise effect should be considered. If the game takes a long time, actions should be optimized to reduce fatigue.

5. Continuous action tips and tutorials;

Detailed tutorials and continuous tips can assist elderly people to better understand and learn actions for game activities. At the beginning of the game, detailed tutorials should be provided, and elderly users should be able to easily obtain help. They should be constantly prompted to input correct information during the game, thus reducing the cognitive load of the interactions and the difficulty of learning and memory. We found that elderly participants were often embarrassed by complex tasks. If reference assistance were provided, this could help them carry out better cognitive activities and show a better acceptance of some metaphorical actions. Additionally, the attention duration of elderly people is shortened, making it difficult to pay attention to a game activity for a long time. To hold their attention, if the game system does not detect interactive behavior, it should prompt the user on various sensory channels.

6. Connection with reality;

Since exergame tasks tend to use common sports activities in real life, the actions in the game should be close to real-world activities to further support active learning and memory. Many older people have no experience with playing games or exercising through video games and often use real-life actions when they initially encounter game tasks related to real activities. Performing game tasks with actions close to reality not only helps elderly people to accept and remember actions but can also enhance immersion in whole-body interactive games. For calling menu and closing menu, the elderly are used to substituting the behavior of operating TV in reality into the tasks; therefore, we suggest using the action of “right index finger clicking an imaginary button in the air”. If the system cannot recognize it or the game interface does not provide virtual buttons for clicking, in order to avoid ambiguity or confusion, we suggest alternative actions: “right hand waving inward” means calling the menu, and “right hand waving left in front of the body” means closing the menu.

7. Reasonable use of body parts;

Appropriate body parts should be used to perform actions, while the feasibility of using other parts instead of performing when the main body parts are occupied should be considered. For example, in the driving game, both hands should be used to perform steering tasks, while the feet should perform tasks such as braking and accelerating. We recommend the action of “stepping on your right foot forward” for braking, and the action of “pushing forward with your elbows at 90 degrees holding the steering wheel” for acceleration task. Designers need to determine the priority of body parts for executing and formulating actions that can be carried out synchronously according to the game scene, making full use of the transferability between hands and legs, as well as between left and right body parts.

8. Make good use of repetitive actions and reversible actions;

The use of repetitive and reversible actions helps reduce users’ psychological load and game operation difficulty. We found that for mirror tasks (turn left/turn right, ski left/ski right, zoom in/out and so on), participants preferred reversible actions. Similarly, in the operation scenarios of system commands, participants preferred repeated pointing actions to perform various operations, which can be distinguished and identified according to simple contextual scenarios. For zoom-in and zoom-out, it is recommended to use the action of “pushing both palms outward to both sides/folding both palms outward from both sides”, which is easy to identify. If the technique of the application scenario is sensitive enough, it is recommended to use the action of “opening after clenching fist with the right hand/pinching the five fingers with the right hand”.

9. Advanced actions;

Advanced actions should be designed for the same game task. We found that factors such as game experience, 3D game experience and realistic experience had significant effects on action preference. To adapt to players with different game backgrounds as the game progresses, advanced actions should be designed for the same game tasks. For the jumping task, it might be recommended that users perform the action of “jump with both feet” at the beginning, with the advanced action of “lift right leg and step down” provided after getting familiar with the game. This not only strengthens the player’s sense of accomplishment but also helps with fatigue management. For jumping, we suggest using the action of “taking off with both feet together” at the beginning of the game and using the action of “lifting your right leg and step down” after becoming familiar with the game, which can not only enhance the player’s sense of accomplishment but also help to manage fatigue.

10. Listen to the advice of rehabilitation experts.

Designers need to consider actions that are feasible in terms of dynamics and ergonomics. They should follow the advice of experts or doctors who specialize in motion control and coordination and pay attention to the assessment of fatigue and physiological risks so that elderly gamers can perform actions easily and smoothly while avoiding uncomfortable movements and postures. For climbing, combined with a physiological risk score, we suggest to use the action of “alternately climbing up with both hands in place”.

### 5.4. Limitations

Due to the current experimental conditions, time, cost and other reasons, the work of this paper is not perfect. There are still some shortcomings and limitations in this research, which need to be further strengthened and improved. For the follow-up research work, there are the following problems and future research directions that need to be considered and paid attention to:

(1) The young subjects in the experiment came from Zhejiang Sci-Tech University, while the old subjects were recruited from the communities around the university, and the proportion of men in the old group was low, which led to the possibility that gender factors might influence the experimental results.

(2) The equipment environment of this experiment does not provide an effective motion recognition system, which is the first choice in the initial stage of the research, because generally using the equipment system requires a certain training foundation and this would also limit the actions that the subjects can put forward. However, this in turn brings some problems; for example, the actions proposed by the subjects may not be recognized in the existing equipment environment, which leads to technical limitations and the increase in development cost. More targeted research may be needed in the future, aiming at a certain full-body interactive game product.

(3) In this study, we only consider the standing interaction using only the whole body, while in the next study, we plan to introduce more body posture interaction, such as studying the whole-body movement design or the whole-body movement with additional equipment when playing driving games in sitting position.

## 6. Conclusions and Future Work

The main results of this study are as follows:

(1) Put forward a set of classification methods of the whole-body interactive movements suitable for exercise games, in which the movements are classified into four dimensions: form, characteristics, position and body parts. On this basis, the differences of action preferences between young people and old people in various dimensions were analyzed. In the formal dimension, the preference of the elderly group for reciprocating motion was higher than that of the young group. On the characteristic dimension, the preference of the elderly group for metaphor and abstract action was higher than that of the young group. In the position dimension, the elderly had a higher preference for environment-independent actions than the young group. No significant difference was found in the dimension of body parts. In addition, in different tasks, the action distribution of the two age groups was also different. Compared with the young group, the old group showed a higher preference tendency of reciprocating, metaphorical abstraction and environment-independent action in more tasks.

(2) By comparing the results of user-defined actions with the results expected by the author, it was found that the actions designed by designers for users are not necessarily actions that conform to users’ cognition, and the importance of bringing users into the design system was emphasized again.

(3) According to the research results and the summarized psychological model of the elderly users, 10 principles of action design of exercise games for the elderly were put forward.

Additionally, we believe that this study provides a reference for other action designs involving or aimed at the whole-body interaction of elderly people. A limitation of this study is that the equipment environment of this experiment does not provide an effective motion recognition system, and we only considered standing-up whole-body interaction. In the next study, we plan to introduce more body posture interaction.

## Figures and Tables

**Figure 1 ijerph-18-12583-f001:**
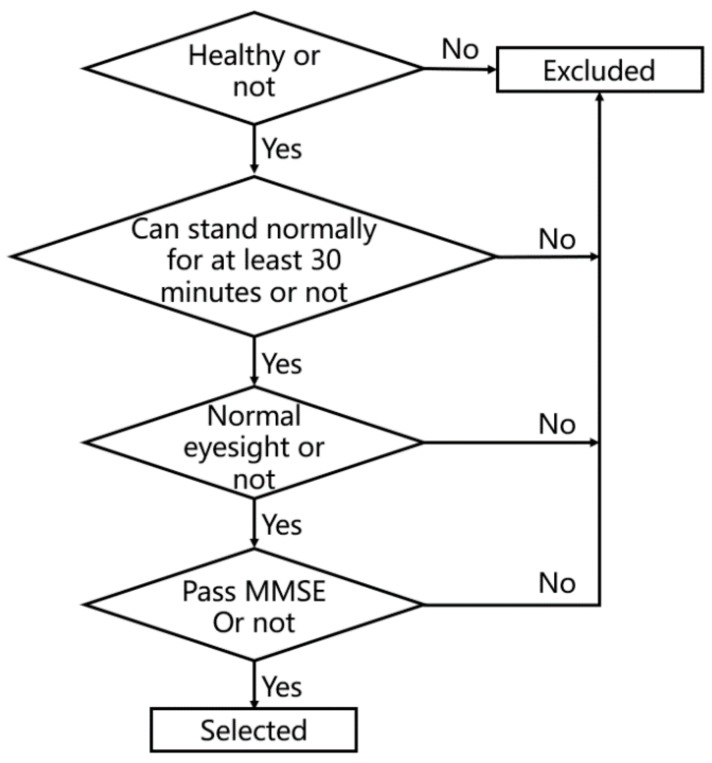
Sample selection.

**Figure 2 ijerph-18-12583-f002:**
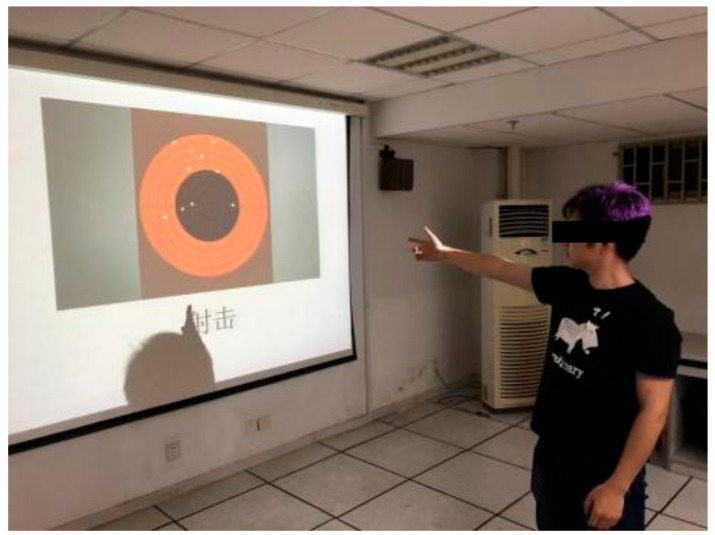
Experimental scene.

**Figure 3 ijerph-18-12583-f003:**
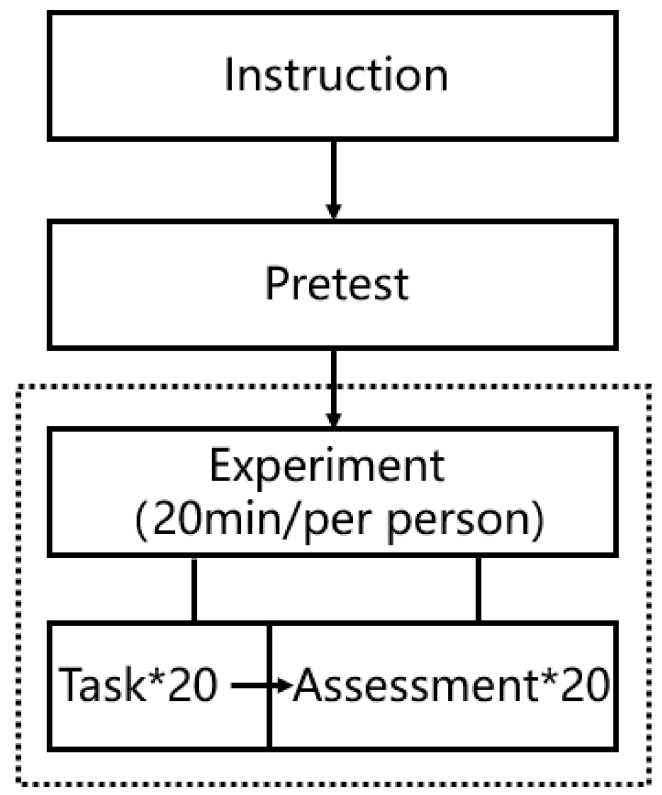
Intervention clarification.

**Figure 4 ijerph-18-12583-f004:**
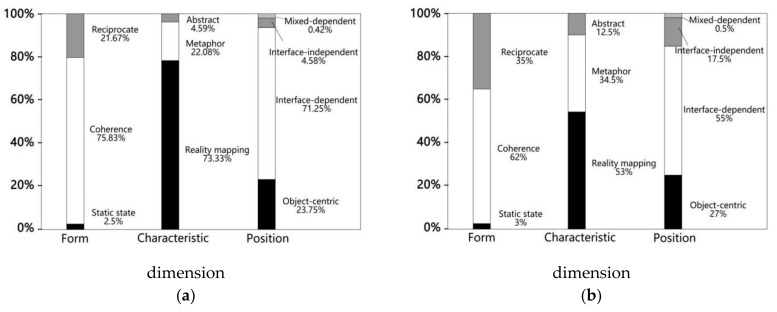
(**a**) Percentage of actions of young group in three dimensions (form, characteristics and position); (**b**) percentage of actions of elderly group in three dimensions (form, characteristics and position).

**Figure 5 ijerph-18-12583-f005:**
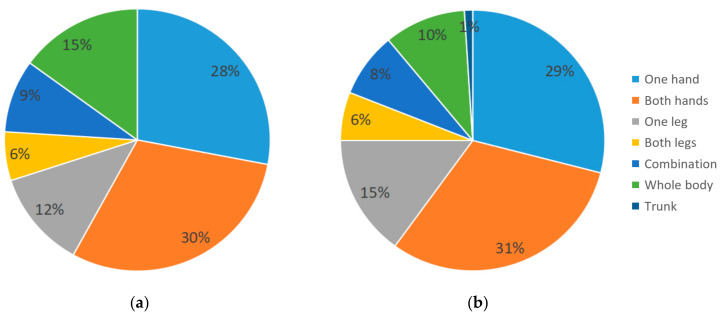
(**a**) Percentage of body parts used by young group, N1 = 240; and (**b**) percentage of body parts used by elderly group, N2 = 200.

**Figure 6 ijerph-18-12583-f006:**
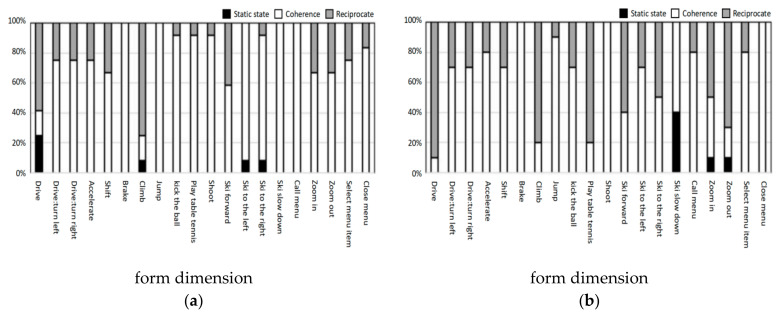
(**a**) Action distribution of young group in the form dimension; (**b**) action distribution of elderly group in the form dimension.

**Figure 7 ijerph-18-12583-f007:**
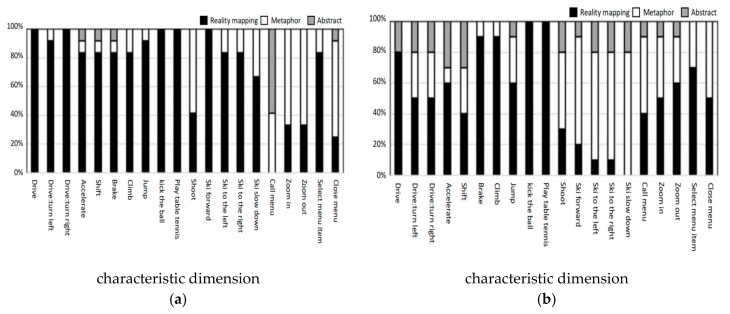
(**a**) Action distribution of young group in the characteristic dimension; (**b**) action distribution of elderly group in the characteristic dimension.

**Figure 8 ijerph-18-12583-f008:**
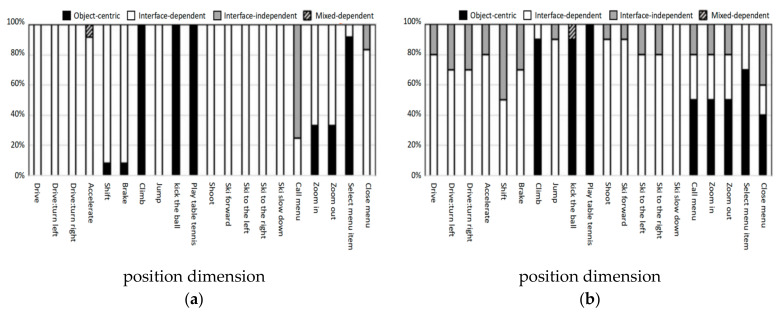
(**a**) Action distribution of young group in the position dimension; (**b**) action distribution of elderly group in the position dimension.

**Figure 9 ijerph-18-12583-f009:**
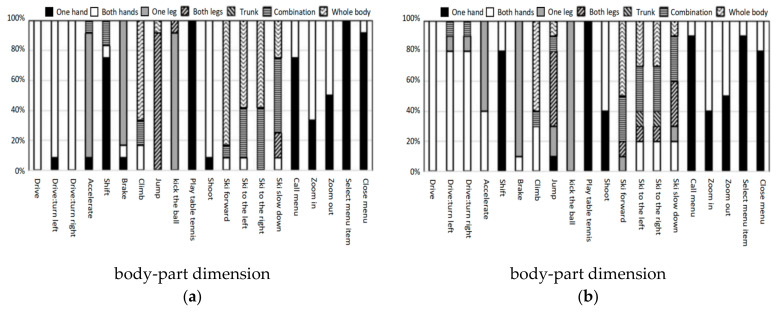
(**a**) Action distribution of young group in the body-part dimension; (**b**) action distribution of elderly group in the body-part dimension.

**Table 1 ijerph-18-12583-t001:** Characteristic information of experimental participants.

Characteristic		Young Group	Elderly Group
Gender	Male	6	2
	Female	6	11
Age		20–30	Over 60
Game experience	More than 15 h a week	7	0
	Less than 15 h a week	5	3
	No	0	10
3D Game experience (Yes)		2	0
Driving experience (Yes)		10	0
Skiing experience (Yes)		2	0

**Table 2 ijerph-18-12583-t002:** Twenty basic tasks.

Category	Task
Driving Game	Drive
	Turn left
	Turn right
	Accelerate
	Shift
	Brake
Sports game	Climb
	Jump
	Kick the ball
	Play table tennis
	Shoot
	Ski forward
	Ski to the right
	Ski to the left
	Ski slow down
System command	Call menu
	Zoom in
	Zoom out
	Select menu item
	Close menu

**Table 3 ijerph-18-12583-t003:** Classification of whole-body interactions.

Dimensions	Categories	Descriptions
Form	Static state	The action is mainly a static posture.
	Coherence	The action includes a coherent movements of body parts.
	Reciprocate	The action includes a multiple reciprocating movements of body parts.
Characteristic	Reality mapping	The action is a mapping of the real world.
	Metaphor	The action expresses metaphorical.
	Abstract	The mapping of the action is arbitrary.
Position	Object-centric	The position is defined for object features.
	Interface-dependent	The position is defined for the environmental features displayed on the interface.
	Interface-independent	The position ignores the environmental features displayed by the interface.
	Mixed-dependent	The position is defined not only for environmental features displayed on the interface but also for object or non-environmental features.
Body part	One hand	The action is mainly performed by one hand.
	Both hands	The action is mainly performed by both hand.
	One leg	The action is mainly performed by one leg.
	Both legs	The action is mainly performed by both leg.
	Trunk	The action is mainly performed by trunk.
	Combination	The action mainly involves two or more body parts but does not produce whole-body movement.
	Whole body	The action involves whole-body movement.

**Table 4 ijerph-18-12583-t004:** Chi-square test results of two groups in different dimensions.

Dimension	Classification	Adjusted Residuals (Young Group)	Sig. (Bilateral)	Cramer’s V Value
Form	Static state	−0.3	0.006 **	0.151
	Coherence	3.1		
	Reciprocate	−0.1		
Characteristic	Reality mapping	4.4	0.000 **	0.221
	Metaphorical	−2.9		
	Abstract	−3.0		
Position	Object-centric	−0.8	0.000 **	0.226
	Interface-dependent	3.5		
	Interface-independent	−4.4		
	Mixed-dependent	−0.1		
Body Part	One hand	−0.3	0.483	—
	Both hands	−0.4		
	One leg	−0.8		
	Both legs	0.2		
	Trunk	−1.6		
	Combination	0.3		
	Whole body	1.7		

**. Significant at 0.01 level (bilateral).

## Data Availability

The data are not publicly available due to the data also forming part of an ongoing study.

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
