# Peer review of "Interaction Preference Differences between Elderly and Younger Exergame Users"

_ijerph, 2021, doi:10.3390/ijerph182312583_

Round 1
Reviewer 1 Report
This study investigates the gesture operation of exergame for elderly and young users, which has not been proposed so far. This study is useful because it discusses the mental models of users and provides design guidelines for exergame gesture operations. The study is well analyzed, and is summarized in detail in the manuscript. For these reasons, this study deserves an Accept.
Reviewer 2 Report
The current paper creates a clear multi-dimensional classification method for whole body interactive actions and proposes a classification method suitable for interactive actions in exergame, which has four dimensions: form, characteristic, position and body part.
Comments to author:
- Please improve the English writing expiration, some of the phrases are very hard to read.
- Please add more details of how the theory from the previous sections is applied is applied in the results section.
- Please add the units of measurement both abscissa and orderly in all figures.
- The authors could add a paragraph with the advantages and the disadvantages of the proposed method.
- The state of the art should be improved with more references, maybe the author could add the following publications:
o Hybrid data-driven fuzzy active disturbance rejection control for tower crane systems, European Journal of Control, vol. 58, pp. 373-387, 2021.
o On model free adaptive control and its stability analysis, IEEE Transactions on Automatic Control, doi 10.1109/TAC.2019.2894586, pp. 1–14, 2019.
Reviewer 3 Report
Thank you for your work. Here are some recommendations:
- The paper is intersting but is necessary to clarify the aim and raise a study hypothesis.
- Add a flow chart to explain the sample selection an a figure to clarify the intervention (i.e. time per session, intensity, volume, replays of the games, attemps, etc.).
- Indicate with subtitles and greater details the statistical analyzes.
- Indicate the results more precisaly (answer to the aim of the study).
- The discussion does not contrast the results of their study with other researchs.
- I think the recommendations should emerge from the resultsof your study in a robust paragraph supported by a figure (own creation). Posing it as a review of the literature is not appropriate and misleads the most relevant aim, wich is the comparison interaction preference differences between elderly and younger people?
- Study limitations should be presented in the discussion.
- The conclusion must be direct and consistent with the aim.
Round 2
Reviewer 2 Report
The authors seriously improved the manuscript of the paper by answering to all my concerns. From my point of view the paper can be accepted to be published in International Journal of Environmental Research and Public Health.
Reviewer 3 Report
The study was substantially improvment, this new version is much clearer. Only, I believe that it is possible to improve the presentation of the results in terms of the significant differences detected, wich would be beneficial for the readers.
